# Eco-Friendly Solution Based on *Rosmarinus officinalis* Hydro-Alcoholic Extract to Prevent Biodeterioration of Cultural Heritage Objects and Buildings

**DOI:** 10.3390/ijms231911463

**Published:** 2022-09-28

**Authors:** Viorica Maria Corbu, Irina Gheorghe-Barbu, Ioana Cristina Marinas, Sorin Marius Avramescu, Ionut Pecete, Elisabeta Irina Geanǎ, Mariana Carmen Chifiriuc

**Affiliations:** 1Department of Genetics, Faculty of Biology, University of Bucharest, 060101 Bucharest, Romania; 2Research Institute, University of Bucharest, 050568 Bucharest, Romania; 3Department of Microbiology and Immunology, Faculty of Biology, University of Bucharest, 060101 Bucharest, Romania; 4Organic Chemistry Department, Faculty of Chemistry, University of Bucharest, 030018 Bucharest, Romania; 5Central Reference Synevo-Medicover Laboratory, 021408 Bucharest, Romania; 6National R&D Institute for Cryogenics and Isotopic Technologies—ICIT, 240050 Ramnicu Valcea, Romania; 7Academy of Romanian Scientists, 050094 Bucharest, Romania; 8The Romanian Academy, Calea Victoriei 25, District 1, 010071 Bucharest, Romania

**Keywords:** microbial biodeterioration, *Rosmarinus officinalis*, vegetal hydro-alcoholic extracts, microbial enzymatic activity, acids, extracellular nitric oxide, antimicrobial, anti-adherence

## Abstract

Biodeterioration of cultural heritage is caused by different organisms capable of inducing complex alteration processes. The present study aimed to evaluate the efficiency of *Rosmarinus officinalis* hydro-alcoholic extract to inhibit the growth of deteriogenic microbial strains. For this, the physico-chemical characterization of the vegetal extract by UHPLC–MS/MS, its antimicrobial and antibiofilm activity on a representative number of biodeteriogenic microbial strains, as well as the antioxidant activity determined by DPPH, CUPRAC, FRAP, TEAC methods, were performed. The extract had a total phenol content of 15.62 ± 0.97 mg GAE/mL of which approximately 8.53% were flavonoids. The polyphenolic profile included carnosic acid, carnosol, rosmarinic acid and hesperidin as major components. The extract exhibited good and wide spectrum antimicrobial activity, with low MIC (minimal inhibitory concentration) values against fungal strains such as *Aspergillus clavatus* (MIC = 1.2 mg/mL) and bacterial strains such as *Arthrobacter globiformis* (MIC = 0.78 mg/mL) or *Bacillus cereus* (MIC = 1.56 mg/mL). The rosemary extract inhibited the adherence capacity to the inert substrate of *Penicillium chrysogenum* strains isolated from wooden objects or textiles and *B. thuringiensis* strains. A potential mechanism of *R. officinalis* antimicrobial activity could be represented by the release of nitric oxide (NO), a universal signalling molecule for stress management. Moreover, the treatment of microbial cultures with subinhibitory concentrations has modulated the production of microbial enzymes and organic acids involved in biodeterioration, with the effect depending on the studied microbial strain, isolation source and the tested soluble factor. This paper reports for the first time the potential of *R. officinalis* hydro-alcoholic extract for the development of eco-friendly solutions dedicated to the conservation/safeguarding of tangible cultural heritage.

## 1. Introduction

Cultural heritage includes tangible heritage assets (buildings, monuments, books, material and decorative objects, natural heritage), and intangible heritage assets, such as language arts, rituals, folklore, oral traditions, and knowledge [1]. The biodeterioration of cultural heritage objects is determined by different biological systems capable of inducing complex alteration processes caused by their biological and metabolic activity, the type of constituent materials of art works and the environmental conditions. In order to inhibit or eradicate biological colonization, chemical methods are usually used in combination with physical ones. However, chemical compounds are generally toxic and non-degradable, persistent in the environment and could cause uncontrollable contamination [2,3]. Thus, over the last decades, trends in control of the biodeterioration process have indicated an increased necessity to develop non-toxic preservation and conservation solutions without adverse effects on cultural heritage objects, human health, and the natural environment [4].

Since ancient times, natural compounds have been used as alternative solutions in the treatment of infectious diseases, although their antimicrobial properties and mechanisms have been only recently proved scientifically. This has prompted the research of green biocides, such as different vegetal natural compounds and essential oils (EO), towards new applications beyond the clinical field, which, due to the biocides microbicidal, antioxidant and anti-biofilm properties, might include conservation and preservation of cultural heritage [5].

Among the vegetal biocides, EOs are among the most studied. This is because they contain different secondary metabolites able to inhibit the growth of bacterial, yeast or filamentous fungi by inducing the deterioration of the cellular walls and plasma membrane, and by interfering with microbial metabolism by affecting the protein synthesis or the structure and function of macromolecules, such as DNA, RNA, proteins and lipids [6,7]. Many plant extracts and essential oils have been described as being alternative solutions for heritage protection against biodeterioration. Although these solutions have a high antimicrobial efficiency it is important that they do not degrade the heritage object itself. Matusiak et al. [8] have described cinnamon essential oil as being safe for heritage textile disinfection since it had a limited impact on the optical, mechanical and structural properties that were observed. Lavin et al. [9] showed that *Origanum vulgare* L. and *Thymus vulgaris* L. EOs exhibited antifungal activity against *Scopulariopsis* spp. and *Fusarium* spp. deteriogenic strains, recommending these EOs for the protection of paper heritage objects against aesthetic and structural damage. *Rosmarinus officinalis* L. is recognized as a medicinal and aromatic plant from the Lamiaceae family native to the Mediterranean region [10,11,12]. EOs extracted from *R. officinalis* L. showed a higher effectiveness against fungal sporulation and growth than, e.g., the commercial biocide benzalkonium chloride [13]. Besides antimicrobial effects, the preventive conservation agents must ensure the reproducibility of the product’s functionality and the optimal potential for inhibiting oxidative degradation. In preventive conservation, the focus is on the environment, mainly temperature, relative humidity and pollutants, which, if not controlled, could accelerate natural decay processes [14]. Thus, antioxidant agents can prevent the oxidation of substrates (formation of side groups, aldehydes and ketones) that may lead to the destabilization of the structure and thus, to easier hydrolysis. Free radicals generated by oxidation or light can contribute to the increase in acid concentration, because by photooxidation, primary alcohol groups in polymers can be oxidized to aldehydes and later to carboxylic acid species [15].

Due to their extremely erosive capacity, fungi are one of the most important biodeteriogenic organisms associated with stone buildings. Black fungi can cause biopitting, which appear as black spots and are resistant to antimicrobial treatments and biocides, owing to their thick walls [16]. Artworks from museums are one of the most affected heritage patrimony objects, with several species of fungi being associated with their biodeterioration. Some genera and species have been reported in association with various types of material and objects from art museum, such as: *Penicillium* spp. (parchment, keratinous substrates, paintings, papers), *Alternaria* spp., *Aspergillus* spp., *Chaetomium* spp., *Mucor* spp. (keratinous substrates, paintings, papers), *Cladosporium* spp. (parchment, keratinous substrates, paintings), *Fusarium* sp. (paintings, paper), *Aureobasidium pullulans* (paintings, keratinous substrates), *Epicoccum nigrum* (parchment, keratinous substrate), *Rhizopus* spp., *Stachybotrys chartarum* and *Trichoderma* spp. (paper, keratinous substrate), while others were isolated from only one type of substrate, such as: *Toxicocladosporium irritans* (paper), *Phlebiopsis gigantea* (parchment) and *Coniosporium* spp. (keratinous substrates) [17]. *Aspergillus* was reported on cultural heritage objects from churches and temples worldwide, the most frequently encountered being *Aspergillus niger* and *A. versicolor* [18]. Among bacterial species involved in biodeterioration *B. subtilis*, *B. megaterium* and *B. mojavensis* are most frequently encountered in Transylvania. Members of these species are highly resistant to nutrient deficiency and other environmental stress conditions exhibiting different characteristics such as the ability to form capsules, spores or biofilm. Most of the biodeteriogenic bacterial strains are able to secrete cellulases, which could be involved in the biodegradation process of wooden objects or esterases, enzymes that affect the murals in stone churches [19].

In Romania there are very few and scarce studies [20,21] regarding the antimicrobial activity of plant extracts against deteriogenic microorganisms of tangible cultural heritage. This study will address this knowledge gap and will present the in vitro antimicrobial activity of an eco-friendly strategy based on *R. officinalis* plant extract, against deteriogenic fungal and bacterial strains. *R. officinalis* is known for its antibacterial, antifungal cytotoxic, anti-rheumatic, antimutagenic, antioxidant, inflammatory, analgesic, astringent carminative, antithrombotic and chemo-preventive properties, which are mediated by different bioactive compounds such as 1,8-cineol, camphor, α-pinene, limonene, camphene, linalool, borneol, rosmarinic acid, caffeic acid, ursolic acid, betulinic acid, carnosic acid and carnosol [22,23,24,25].

The present study aimed to evaluate the capacity of the hydro-alcoholic extract of *R. officinalis* to inhibit the growth and to modulate the production of soluble factors of deteriogenic fungal and bacterial strains previously isolated from the cultural heritage museum collections and churches in Romania. For this, the physico-chemical characterization of the extract, it’s antimicrobial, antibiofilm and antioxidant activity, as well as it’s modulatory activity on the production of soluble factors involved in biodegradation were demonstrated on a representative number of filamentous fungi and bacterial strains isolated from biodeteriorated wooden and stone heritage churches from Arad, Hunedoara Bucharest and Tulcea counties and from objects included in museum collections.

## 2. Results

### 2.1. Chemical Composition and Antioxidant Activity

The obtained extract has a total phenol content of 15.62 ± 0.97 mg GAE/mL which approximately 8.53% were flavonoids (Table 1). The polyphenolic profile of this extract is characterized by the presence of carnosic acid, carnosol, rosmarinic acid and hesperidin, as major components [26]. HPLC-MS identified approximately 2.04% of the phenolic content and 14.19% of the total flavonoids (Table 2). Among the compounds identified by RP-HPLC-MS, the highest concentrations were obtained for hesperidin (10.39 mg/L), vanillic acid (7.51 mg/L) and quercetin (7.10 mg/mL). By MS, the compounds carnosol and rosmarinic acid were identified, the latter being the one of the main compounds. The main compound from *R. officinalis* extract, previously known for its broad antimicrobial spectrum, could be considered the rosmarinic acid (Appendix A). 

The best antioxidant activity for *R. officinalis* extract was obtained by the TEAC method followed by CUPRAC (Table 1), probably due to the similarity of redox potential between the redox couple Cu (II, I) –Nc (0.6 V) and ABTS+/ABTS (0.68 V) [27]. Another favourable reason to use TEAC and CUPRAC is that these methods are able to measure both hydrophilic and lipophilic antioxidants as opposed to the FRAP and DPPH methods [28]. In addition, the TEAC method can measure the antioxidant activity over a wide pH range, and the increase in acidic conditions due to the high content of phenolic acids facilitates the electron transfer [29]. The considered positive control was ferulic acid. This compound was found in the extract composition, the proportion of its antioxidant activity can be found in Table 1. The antioxidant activity of the extract is given by the synergisms/antagonisms between the amounts of compounds.

The total ion current (TIC) chromatogram of *R. officinalis* extract by UHPLC–MS/MS detection is presented in Appendix A. Quantitative UHPLC–ESI/MS results indicate that the main phenolic acids identified in the alcoholic extract of *R. officinalis* were vanillic acid (7512.58 µg/L), caffeic acid (1637.63), chlorogenic acid (999.51µg/L), and syringic acid (836.12), while hesperidin (10,389.09 µg/L) and epi-catechin (664.45 µg/L) were the predominant flavonoids (Table 2). The identification of the main quantified phenolic acids and flavonoids are presented in Appendix A.

The analytical approach based on a non-target UHPLC-Q-Orbitrap HRMS analysis allows the identification of other bioactive compounds and specialized metabolites which are also responsible for the bioactive potential of *R. officinalis* extract. Data processing analysis using Compound Discoverer software following a metabolomics working template includes RT alignment, molecular formula prediction, evaluation of adducts, the assignment and comparison of fragmentation pattern, background annotation, and an automated library and database search for identification purposes, including mzCloud (MS2 fragments), Chemspider, MzVault and Mass List Matches [30].

The most represented classes of bioactive compounds that were identified in *R. officinalis* extract are terpenoids (di- and tri-terpenoids) and flavonoids (flavones, flavanones, and flavonols). The compounds name, molecular formula, retention time, exact mass and accurate mass of m/z adduct ions, and MS/MS fragment ions in negative HESI mode are shown in Table 3. Diterpenoids compounds, such rosmanol, rosmaquinone and sophorol isomers, carnosol and carnosol derivatives, with strong anti-cancer and anti-inflammatory properties [31] were identified as predominant compounds in the rosemary extract (Appendix A). Oleanolic and ursolic triterpenic acids, typical constituents in medicinal plants, with antibacterial, anti-inflammatory and anti-proliferative [32] activities, were also identified. Among flavonoids, flavones represent the predominant group, being present as glycosylated forms of luteolin, apigenin, isorhamnetin, hispidulin, and also dihydroxy-dimethoxy flavone, compounds with strong antioxidant and anti-proliferative activities [33]. Phenolic acids such as rosmarinic acid and methyl rosmarinate derivate and usnic acid were also identified.

### 2.2. Qualitative Screening of the Antimicrobial Activity

#### 2.2.1. Antifungal Activity

A total number of 41 filamentous fungi strains isolated from different wooden and stone churches and from museum objects (paper and paintings) were identified according to presumptive examination and MALDI-TOF mass spectrometry as belonging to *A. niger*, *A. flavus*, *A. pseudoglaucus*, *A. nidulans*, *A. flavus oryzae*, *A. versicolor*, *A. montevidensis*, *A. clavatus*, *P. chrysogenum*, *P. corylophylum*, *P. expansum*, *P. digitatum*, *P. brevicompactum*, *P. namylowskii*, *Rhizopus oryzae*, *A. alternata*, *Fusarium proliferatum*, *Mucor circinelloides*, *Purpureocillium lilacinum* species [19].

The antifungal activity of *R. officinalis* hydro-alcoholic extract against 41 filamentous fungi strains, mainly belonging to *Penicillium* (49.9%) and *Aspergillus* genera (31.70%) was performed by an adapted diffusion method which demonstrated a good inhibitory effect of the extract as compared to solvent (Appendix A). Therefore, the extract inhibited 60.97% of the tested filamentous fungal strains while the solvent only inhibited 21.95% (Figure 1).

#### 2.2.2. Antibacterial Activity

For the antibacterial activity a total number of 15 bacterial strains isolated from wooden and stone churches and from museum collections were selected as representative for the microbial communities developed on cultural heritage objects from Romania [19].

The qualitative screening of the antimicrobial activity of the hydro-alcoholic extract of *R. officinalis* showed that the bioactive compounds inhibited the growth of all tested bacterial strains, evidenced by the total (Appendix A) and increased diameter of the induced growth inhibition zone compared to the solvent (Figure 2).

### 2.3. Quantitative Evaluation of the Antimicrobial Activity

#### 2.3.1. Antifungal Activity

The distribution of the MIC values obtained using the binary serial dilution method by the isolation sources revealed that the most susceptible strains were recovered from the deteriorated textiles (MIC = 25.03 mg/mL) followed by stone and wooden churches, while the most resistant corresponded to other museum objects (MIC = 100 mg/mL) (Figure 3).

The MIC distribution by the most frequently deteriogenic genera of the cultural heritage churches and museum collections (*Penicillium* spp. and *Aspergillus* spp.) (Appendix A) has shown that *R. officinalis* hydro-alcoholic extract demonstrated the highest efficiency against *A. clavatus* (MIC = 1.2 mg/mL) and *Aspergillus* spp. strains (MIC = 6.25 mg/mL), while the *P. namyslowskii* and *P. brevicompactum* were the most resistant (MIC = 100 mg/mL).

#### 2.3.2. Antibacterial Activity

The quantitative evaluation of the antimicrobial efficiency of *R. officinalis* hydro-alcoholic extract against bacterial strains recovered from different sources revealed that the highest efficiency was detected for strains isolated from paintings and textile objects (MIC = 1.56 mg/mL) (Figure 4). Significantly lower MIC values (*p* < 0.01), except for strains isolated from textiles and paintings, were obtained for the extract compared to the solvent, demonstrating that the antibacterial activity was mainly due to the bioactive compounds contained by *R. officinialis* extract.

The MIC value ranges within each isolation source probably due to different physiological characteristics of the bacterial strains isolated from the respective source (Figure 4).

Concerning the distribution of MIC values by bacterial species, the most susceptible strains belonged to *Arthrobacter globiformis* (MIC = 0.78 mg/mL), followed by *Bacillus cereus*, *B. thuringiensis* and *A. aurascens* (MIC = 1.56 mg/mL) and the most resistant corresponded to *Pseudomonas koreensis* recovered from stone churches (MIC = 6.25 mg/mL) (Appendix A). The same significantly lower MIC values were recorded for the *R. officinialis* extract as compared to the solvent (*p* < 0.01), except for *B. atrophaeus* and *A. aurascens* strains, suggesting the large spectrum antibacterial activity of the bioactive compounds contained by the tested extract.

### 2.4. The Inhibition of Microbial Adherence to the Inert Substratum

To determine the potential of *R. officinalis* extract to inhibit the ability of fungal and bacterial strains to adhere to the inert substrate, we selected six fungal and three bacterial strains which, according to previous studies, have shown a high capacity of biofilm development on the surface of heritage objects [19]. For this purpose, the extract and the solvent were diluted to final concentrations corresponding to MIC, MIC/2 and MIC/4.

The *R. officinalis* extract with a concentration equivalent to the MIC value or concentrations lower than the MIC value has also been able to inhibit the adhesion to the inert substrate of the most tested fungal strains (Figure 5). Apart from the *P. chrysogenum* S2042 strain, in most of the cases, different concentrations of the extract proved to be significantly more efficient than the solvent (*p* < 0.05). The best results were obtained against *P. chrysogenum* MTR3 and S1787strains isolated from wooden (*p* < 0.001) or textiles objects (*p* < 0.01) for all tested concentrations (Figure 5).

According to Figure 6, the chemical compounds from the composition of *R. officinalis* have also inhibited by different degrees the biofilm developed by the three selected bacterial strains, as revealed by the low MBEC values, the anti-biofilm effect depending on the bacterial species and the tested concentration. This might possibly be explained by the different composition of biofilm matrix formed by different bacterial strains, influencing the rate of anti-biofilm compounds diffusion inside the bacterial biofilm. Thus, for *B. pumilus*, the best anti-biofilm activity was recorded at MIC/2, followed by MIC/4, for *B. thuringiensis* at the MIC value and for *P. koreensis*, all tested concentrations exhibited an anti-adherence effect, although not statistically significant as compared to the solvent.

The anti-biofilm effect seems to correlate, at least in some cases with the inhibition of the NO release (Table 4).

### 2.5. Effect of R. officinalis Hydro-Alcoholic Extract on the Enzymatic Activity

From the tested fungal strains, 32% produced cellulases, 83.33% caseinases, 78.57% amylases and 42.85% produced organic acids. None of the tested deteriogenic strains produced phenoloxidases. All tested bacterial strains produced amylases, 88.88% caseinases, 38.88% esterases and none produced cellulases.

Starting from these results, strains producing different soluble factors were further selected to assess the influence of the rosemary extract on the production of each of them. Regarding the production of cellulases, no significant inhibition was noticed for any of the tested strains. Regarding caseinases, the enzymatic production was significantly inhibited by the extract in comparison to solvent in the cases of the *P. chrysogenum* strain isolated from textiles and the *A. nidulans* strain isolated from stone churches (*p* < 0.05). The production of amylase was significantly inhibited in the case of *A. nidulans* strain from stone churches and of *A. montevidensis* from textiles (Table 5 (a)).

Thus, the impact of the extract on the biodegradative enzymes production differs depending on the studied strain. 

In the case of the five fungal strains producing organic acids, when they are cultivated in the presence of *R. officinalis* extract, the production and secretion capacity of these types of compounds increases (the obtained values being higher than 100%). However, the stimulating role seems to be due to the solvent used for extraction and not to the extract itself. This conclusion is sustained by the fact that, in the case of the *P. chrysogenum* strain isolated from wooden churches, the presence of the extract diminishes the stimulating effect of the solvent from 147.62% to 104.76% (*p* < 0.001) (Table 5 (b); Appendix A).

The impact of *R. officinalis* extract on the enzymatic activity was tested on three bacterial strains belonging to the genus *Bacillus*. The presence of *R. officinalis* extract does not significantly influence the ability of bacterial strains to produce caseinases or amylases. The general effect was the stimulation of the production of these enzymes but the effect is rather due to the solvent, excepting the *B. cereus* strain isolated from a museum object, for which the presence of chemical components in the structure of the extract decreases the stimulatory effect of the solvent (*p* < 0.05) (Table 5 (a)).

## 3. Discussion

The present study is part of an important current research direction of our group, aiming at the development of effective strategies to combat deteriogenic microorganisms and preserve cultural heritage objects in Romania. According to the latest research on the restoration and conservation of heritage objects, the use of biocides based on chemicals of natural origin, to the detriment of biocides obtained by chemical synthesis, has a number of advantages, e.g., diminished risk of side effects for human and environmental health [34]. Several studies have shown that naturally occurring compounds have antimicrobial and antioxidant activities that depend on several factors: organic/inorganic artifacts; indoor/outdoor environments; bacterial or fungal colonizers; the biological activity of different solutions; and the mechanisms of action [35]. Most of these have focused on the development of biocides based on EOs [36,37] and less on hydro-alcoholic plant extracts. Even so, it has been proven that various extracts derived from plants such as *A. sativum* [21], *Clinopodium nepeta* [38] or *Robinia pseudoacacia* [39] can be efficient for treating different cultural heritage objects.

In this context, our study aimed to demonstrate and characterize the efficiency of *R. officinalis* hydro-alcoholic extract to inhibit the growth of deteriogenic fungal and bacterial strains previously isolated from cultural heritage objects and buildings from Romania. In order to achieve the proposed objective, the first stage consisted in obtaining the hydro alcoholic extract. The chosen extraction method was based on the study conducted by Švarc-Gajić et al., in 2013 [40], where the polyphenol content differs depending on the solvent used. Higher phenol yields were obtained by using methanol, which is the most widely used solvent in the literature [41,42] but with a strong toxic effect, so it is not recommended to use [43]. The lowest extraction yield was obtained for ethyl acetate, even if the dipole moment is not very different from methanol (1.78 and 1.69, respectively) [40]. This effect can be explained by the much higher dielectric loss coefficients observed in methanol than in ethyl acetate [44]. The next high phenol content was obtained using ethanol; thus, we have selected this solvent and the extraction was finally made in 50% ethanol.

The next step was to evaluate the antimicrobial efficacy of the obtained extract against 41 fungal strains and 15 bacterial strains isolated from the surface of cultural heritage objects and buildings in Romania. The antimicrobial activity of *R. officinalis* extract differs depending on the isolation substrate and the microbial species. The best results were obtained for Gram-positive bacterial strains such as *A. globiformis* (MIC = 0.78 mg/mL) or *B. cereus*, *B. thuringiensis* and *A. aurascens* (MIC = 1.56 mg/mL), most likely due to the chemical composition of the cell wall and membrane, while in the case of fungi the most sensitive strains belong to *A. clavatus* species (MIC = 1.2 mg/mL). Further, a study confirmed that *R. officinalis* extracts are more efficient against Gram-positive compared to Gram-negative bacteria, mostly due to the presence of carnosic acid, carnosol and rosmarinic acid [45].

The antimicrobial activity of *R. officinalis* hydro-alcoholic extract against both fungal and bacterial strains seem to be correlated with the presence of hesperidin (10.39 mg/L), vanillic acid (7.51 mg/L) and quercetin (7.10 mg/mL). These compounds have a large spectrum of antimicrobial activity; for example, in *Citrus* spp., hesperidin was considered the most active compound against *Staphylococcus aureus*, *Escherichia coli* and *B. subtilis* [46].

Hesperidin is a strong antifungal and is active against *A. parasiticus*, *A. flavus*, *F. semitectum* and *P. expansum* [47]. Vanillic acid exhibits antimicrobial activity against *Enterobacter cloacae* and *E. hormaechei* strains; further, it induces a delay in the development of *Trichoderma* spp., *Athelia rolfsii*, *Phaeoacremonium parasiticum*, *A. niger* and *F. oxysporum* [48,49,50]. The *R. officinalis* extract also contains other derivatives of caffeic acid, that could act as ligands for metal ions; thus, they react with peroxide radicals and stabilize free radicals. The catechol group is responsible for radical electron capturing formed as a result of oxidation. The skeleton formed by the three rings allows the charge relocation. The presence of the carboxylic group increases this conjugation, especially in aqueous systems. However, in slightly polar environments, such as fats, the structure of lactone seems to provide greater stability [51].

Although a limited number of studies have focused on demonstrating the efficiency of *R. officinalis* extract against biodeteriogenic microorganisms, the antifungal activity of *R. officinalis* extracts (water and ethanol) against four fungal strains (*A. flavus*, *A. versicolor*, *Penicillium* spp., *P. purpurogenum*) isolated from different archaeological artefacts in an Egyptian Museum demonstrated the efficiency of both extracts [52]. Several other authors have demonstrated the antimicrobial activity of *R. officinalis* EO or of the crude extract against several food-borne pathogens, for i.e., against *Macrophphaseolina phaseolina* to control charcoal rot in *Glycine max* [53]. The *R. officinalis* showed strong antifungal activity by its disruption of the cell wall and the loss of cellular components and by its inhibition of fumonisin B1 and B2 production in a *F. verticillioides* Sacc. strain previously isolated from corn residue used in animal feed (MIC = 150 µg/mL) [54]. 

The role of antioxidants in an artifact’s preservation depends on their physico-chemical nature and possible oxidation reactions that may occur on the surface depending on environmental conditions. In the case of woody artifacts, it was found that the initial stage of wood degradation by fungi occurs through oxidative reactions. Free radical species are involved in the initial stage of wood degradation by fungi. To increase the pore size that facilitates the penetration of extracellular enzymes, the wood treated with an antioxidant becomes more resistant to damages caused by cellulase-producing fungi [55]. In the case of paintings and textiles, it has been found that with the passage of time the flavonoids from the paint’s compositions are transformed into phenolic acids, thus suffering an oxidative degradation. Thus, the use of natural antioxidants could contribute to a better conservation of these types of cultural heritage [56].

The establishment of surface-associated assemblages of microorganisms on different heritage objects is an essential step in the biodeteriogenic process, providing them enhanced resistance to physical and chemical stress [57]. Thus, a better understanding of the processes that allow cells to attach to different surfaces could allow the clarification of the relevance of biofilm formation and subsequently induced biodeterioration. Although it is of great importance to develop biocidal solutions for the decontamination of heritage objects, the prevention of secondary contamination brings a special economic advantage. For this reason, the *R. officinalis* extract has also been tested as a solution to prevent the adherence of deteriogenic microbial strains to the inert substrate. Our study reports for the first time in the literature the anti-adherent effect of *R. officinalis* hydroalcoholic extract against biodeteriogenic strains. The results show that the anti-adherence effect of *R. officinalis* extract occurs at different concentrations depending on the tested fungal or bacterial strain. No specific correlation with the isolation source or species could be noticed. However, the *R. officinalis* extract has shown a good anti-adherent activity against *P. chrysogenum* strains isolated from wooden or textile materials or *B. thuringiensis* strains.

The high efficiency of *R. officinalis* extract against oral planktonic bacteria recommended it as an adjunct in the treatment and prevention of biofilms in the oral cavity [58]. Further, hesperidin impedes the adherence of *S. aureus* to inert substratum by a molecular mechanism involving the down regulation of genes such as: polysaccharide intracellular adhesion gene (icaA and icaD), fibronectin-binding protein coding gene (fnbA and fnbB) and staphyloxanthin production (crtM) [59]. Vanillic acid also acts at the molecular level, reducing the expression of genes involved in biofilm development in the case of *Vibrio alginolyticus* strains [60].

Intracellular reactive species play critical roles in microbial vitality, including stress management and differentiation [61]. Among all reactive species, NO has been considered a universal signaling molecule in most organisms, including microorganisms. However, for fungi, the synthesis and functions of NO have not been fully elucidated. In most fungi, endogenous NO generation has been indirectly revealed by identifying NOS activity or using NO absorbers. Few studies have directly shown the presence of intracellular NO [62,63,64].

It has been found that oxidative and nitrosative stress could occur inside biofilms, affecting their development and maturation under various conditions [65]. The production of ROS and RNI and accumulation of these radical oxidants in the extracellular environment could affect the biofilm matrix. Low, non-toxic endogenous NO concentrations, could also induce the dispersion of the biofilm [66]. In bacteria, endogenous NO can be generated from L-arginine by bacterial NOS. Alternatively, NO can be produced by NOS-independent pathways, such as the reduction of nitrites to NO by nitrite reductases in denitrifying bacteria such as *Bacillus* spp., *Serratia* spp. and *Pseudomonas* spp. [67]. In the case of bacteria, the production of NO is closely associated with the formation and dispersion of bacterial biofilm [66]. Low NO concentrations (25 pM to 2.5 μM NO) inhibited biofilm formation, those above 2.5 μM NO, promoted biofilm and concentrations greater than 1 mM NO had a bactericidal effect [68]. Low NO concentrations regulate a widely conserved genetic pathway that controls the conversion between planktonic and sessile lifestyle through the action of c-di-GMP [69]. A consequence of NO-mediated dispersal is that, due to the reversal of the genetic program, both the biofilm and the dispersed cells lose their antimicrobial resistance [70]. The effect of NO on the inhibition and dispersal of biofilm has been studied in both Gram-negative and Gram-positive bacteria, such as *Serratia marcescens*, *Vibrio cholerae*, *E. coli*, *Fusobacterium nucleatum*, *B. licheniformis* and *S. epidermidis*.

The method proposed in our study allows quantification of the NO content in a liquid medium without distinguishing between specific sites or structures in which the NO is produced or accumulated. Endogenous NO has been detected in fungi during the formation of conidia on solid medium and the branching of hyphae in liquid medium by a fluorescence method [62]. In addition, high fluorescence detected inside vacuole-like structures in fungal hyphae may indicate the subcellular site of NO synthesis or may be the result of fluorogenic compound accumulation in vacuoles. Moreover, although the highest NO level was observed during the formation of microconidia, fluorescence was largely detected in vegetative hyphae, not in phialides and microconidia. Vacuoles, in general, act as a reservoir for the storage of basic amino acids, such as arginine, in filamentous fungi [71]. Thus, NOS-dependent NO synthesis may be responsible for the increased NO concentration.

The successive colonization of heritage objects and the action of secondary metabolites produced by biodeteriogenic microorganisms lead to the loss of the quality of heritage objects, affecting both their integrity and appearance. The main enzymes involved in biodeterioration are esterases, lipases, proteases, cellulases, glucanases or laccase. Depending on the material from which the heritage objects are made, the relevance of the enzymes involved in biodeterioration is different. For paintings or lacquered church objects, esterases or lipases are the most aggressive. Meanwhile, for wooden or paper objects [72], cellulases and proteases have the most harmful effect [72,73]. Similarly, in the case of stone objects or derivatives the organic acids have a significant impact [74].

The microbial strains included in this study presented various characteristics regarding the production of enzymes involved in the biodeterioration. Thus, for each type of enzyme, several representative microbial strains were selected to determine whether *R. officinalis* extract influences their ability to produce and secrete the enzyme of interest in the culture medium. According to the obtained results, the treatment of microbial cultures with sub inhibitory concentrations of extract or solvent leads either to the inhibition of the enzymatic activity or to its stimulation, the results being different depending on the studied strain, the type of tested enzyme or the isolation source. From a statistical point of view, in most situations it is observed that the solvent is that which determines the modification of the enzymatic activity. There are few exceptions to this, in which the tested extract inhibited the capacity of *A. nidulans* (stone church) to secrete amylases or caseinases, of *A. montevidensis* from textiles to secrete amylases, of *P. chrysogenum* isolated wooden church to produce organic acids or of the *B. cereus* strain isolated from a museum object to produce esterases.

Although not tested in biodeterioration studies, chlorogenic acid, an important compound present in the *R. officinalis* extract has been shown to inhibit carbohydrates-hydrolyzing enzymes such as amylases and glucosidases being currently investigated as hypoglycaemic adjuvant [75]. Other phenolic compounds such as p-coumaric acid, vanillic acid, syringing acid, benzoic acid, and ferulic acid are known for their ability to inhibit the production of enzymes such as laccase (that might be involved in textile discoloration), as demonstrated for *Botryosphaeria* genera [76,77]. Further, it has been shown that caffeic acid, p-coumaric acid, and ferulic inhibit protease, cellulase and esterase production in *F. oxysporum,* while p-coumaric acid and ferulic acid strongly decrease cellulase activity in the case of *T. reesei* strains [78,79]. Similarly, rutin, quercetin, apigenin or kaempherol inhibit cellulase production in *F. solani* [80]. Green alternatives used for heritage protection may refer to the prevention of microbial contamination or the development of control strategies that are based only on inhibiting maturation or dispersal of biodeteriogenic microbial strains [34]. Microbial contamination prevention can be achieved either by treating the affected area with specific biocidal compounds or by exposing the heritage objects to inhibitory vapours existing in the repository environment. For example, essential oils with a high content of volatile antimicrobial compounds can be used to prevent biodeterioration without being directly applied to the surface of the affected cultural heritage objects [81]; however, exposure to the vapours can cause significant damage to the structure of the heritage object. Such a situation has been described for thymol, the main chemical component of thyme essential oil which was used frequently in the past in so-called “thymol cabinets”. Although it has great biomedical potential (having antimicrobial, anti-inflammatory or antioxidant properties) [82] it cannot be used for heritage protection since it softens the varnishes and resins, renders parchment brittle and degrades paper objects as well iron gall ink or watercolour binders [34,83]. Therefore, an important step for describing new biocidal solutions to be used for biodeterioration control or prevention is to determine the threshold above which the solution is effective against microbial strains but at the same time does not imply any risk for the object itself. Regarding rosemary extract, there is a limited number of studies concerning its possible use as an ecological alternative for heritage protection. Even so, Di Vito et al., have stipulated that the rosemary extract can be included in gellan hydrogel to obtain a cleaning agent for paper artwork with no negative impact on heritage object structure or integrity [84]. Taking into account that the antimicrobial efficiency of the plant extract proven by this work is sufficiently high and that the minimum concentration of plant extract necessary to inhibit the growth of microbial strains, their adhesion to the substrate and their enzymatic activity being relatively low, it can be recommended to proceed further with future studies for testing the effectiveness of rosemary extract or polymeric matrices containing rosemary extract as antimicrobial agents. Further studies should also include their impact on different substratum models such as exterior deteriorating wood, stone, mural paintings or paper by SEM, ATR-FTIR or gravimetrical methods.

## 4. Materials and Methods

### 4.1. Microwave Assisted Extraction (MAE)

A microwave-assisted extraction (MAE) procedure was carried out for the extraction of *R. officinalis* bioactive compounds using a microwave extractor (Ethos SEL, Milestone, Brondby, Denmark). For extraction, plant material powder was mixed with selected solvents (ethanol: ultrapure water = 50:50) at a ratio of 1:10 (*w*/*v*) and underwent MAE for 1 h at 100 °C. After the MAE procedure, the *R. officinalis* extract was filtered with Whatman filter paper No.1, collected in a volumetric flask and stored refrigerated at 4 °C until further use.

### 4.2. Chemical Analysis of R. officinalis Extract

#### 4.2.1. Total Phenolic Content (TPC)

TPC was performed according to Singleton et al., 1999 [25]. To 10 µL of sample or gallic acid solutions were added 90 µL H_2_O, 10 µL of Folin Ciocalteu reagent, 100 µL of Na_2_CO_3_ 7% and 50 µL of H_2_O. After 60 min of incubation at room temperature and in the dark, the absorbance was measured at 765 nm. The calibration curve was done for concentration between 25–250 μg/mL of gallic acid (R^2^ = 0.9994).

#### 4.2.2. Total Flavonoids Content (TFC)

TFC was determined by the aluminium chloride method [85]. Specifically, 0.1 mL sample/standard with 0.1 mL of 10% sodium acetate. An amount of 0.12 mL of 2.5% AlCl_3_ solution was added to this mixture and made up to a final volume of 1 mL with 70% ethanol. After stirring, this solution was left for 45 min in the dark and the absorbance of the mixture was read at 430 nm. The calibration curve was performed for quercetin concentrations in the range of 10–250 μg/mL (R^2^ = 0.9968).

#### 4.2.3. Determination of Reducing Sugar Content (RSC)

RSC was determined using the 3,5-dinitrosalicylic acid (DNSA) method, as previously described with slight modification [86]. For the measurement, 330 µL of DNSA reagent was pipetted into a test tube containing 170 µL of plant extract and kept at 90 °C for 15 min. After cooling, a volume of 150 μL was pipetted from each reaction mixture in 96-well plates and the absorbance of the resulting solution was measured at 540 nm using a UV-VIS spectrophotometer. The reducing sugar content was calculated from the calibration curve of standard D-glucose (0.1–1 mg/mL) (R^2^ = 0.9972).

### 4.3. Characterization of the Extract by UHPLC–MS/MS

Polyphenolic compounds quantification was performed using a high-resolution Q Exactive mass spectrometer™ Focus Hybrid Quadrupole–OrbiTrap (Thermo Fisher Scientific, Waltham, MA, USA) equipped with HESI, coupled to a high-performance liquid chromatograph UltiMate 3000 UHPLC (Thermo Fisher Scientific). Chromatographic separation was performed on a Kinetex^®^ C18 column (100 × 2.1 mm, 1.7 µm particle diameter) at 30 °C, under a gradient elution of two mobile phases: A—water with 0.1% formic acid and B—methanol with 0.1% formic acid, at a flow rate between 0.3 and 0.4 mL/min, as presented by [87]. Full scan data in negative mode covering a scan range of m/z 75–1000 was acquired at a resolving power of 70,000 FWHM at m/z 200, while variable data-independent analysis MS2 (vDIA) was performed at the resolution of 35,000, isolation windows and scan ranges being set as follows: 75–205 m/z, 195–305 m/z, 295–405 m/z, 395–505 m/z and 495–1000 m/z. Nitrogen was used as collision gas and auxiliary gas at a flow rate of 11 and 48 arbitrary units, respectively. The applied voltage was 2.5 kV, the capillary temperature was 320 °C and the energy of the collision was set as 30 eV. Polyphenolic compound quantification and calibration was performed in the concentration range between 0 and 1000 μg/L for each of the phenolic acids and flavonoids by serial dilution with methanol of the standard mixture of concentration 10 mg/L. Data processing, analysis and interpretation were performed with Xcalibur software package (Version 4.1) and Compound Discoverer v. 2.1 software (Thermo Fisher Scientific) using an untargeted metabolomics working template.

### 4.4. Antioxidant Activity

The measurement of DPPH assay was performed by a method adapted from [88]. To 100 μL of sample/standard was added 100 μL of 0.3 mM DPPH radical solution. The absorbance was read at λ = 517 nm after 30 min of incubation in the dark. The calibration curve was achieved for concentrations between 80–5 µM Trolox/mL (R^2^ = 0.9988). The results were expressed in mM Trolox equivalent/mL extract.

**The CUPRAC method** is based on the reduction of a cupric complex, neocuproin, by antioxidants in copper form. Copper ion reduction was performed as previously described [89]: 60 µL of sample/standard solutions of different concentrations were mixed with 50 µL CuCl_2_ (10 mM), 50 µL neocuproin (7.5 mM), and 50 µL ammonium acetate buffer 1 M, pH = 7.00. After 30 min, the absorbance was measured at 450 nm. The stock Trolox solutions required for the calibration curve were 2 mM, and the working concentrations were between 0.24 and 2.0 mM Trolox/mL (R^2^ = 0.9991). The results were expressed in mM Trolox equivalent/mL extract.

The determination of the antioxidant capacity of iron reduction was performed by the **FRAP assay method** [90]. The stock solutions included 300 mM acetate buffer, pH 3.6, 10 mM 2,4,6-tripyridyl-s-triazine (TPTZ) solution in 40 mM HCl, and 20 mM FeCl_3_ 6H_2_O solution in a volume ratio of 10:1:2, which were warmed to 37 °C before use. After incubation, the absorbance was read at 593 nm. A 1 mM Trolox stock solution was used to plot the calibration curve, the concentration ranging between 30 and 250 µM Trolox/mL (R^2^ = 0.9971). The results were expressed in mM Trolox equivalent/mL of extract.

**Trolox equivalent antioxidant capacity (TEAC)** assay was performed according to Re et al., [91] with few modifications. A stable stock solution of ABTS+ was produced by mixing a solution of 7 mM ABTS in 2.45 mM potassium persulphate. Then, the mixture was left standing in the dark at room temperature for 12–16 h before use. An ABTS+ working solution was obtained by dilution with ethanol to an absorbance of around 0.70. The reaction mixture consisted in 20 µL of sample/standard and 180 µL of ABTS+ working solution and was incubated 30 min in the dark. The standard curve was linear between 20 and 200 µM Trolox (R^2^ = 0.9981). The results were expressed in mM Trolox equivalent/mL extract.

### 4.5. Antimicrobial Activity

#### 4.5.1. Qualitative Screening of the Antimicrobial Activity of *R. officinalis* Hydro-Alcoholic Extract

An adapted diffusion method using 10 μL of hydro-alcoholic extract (50:50; 100 mg/mL) was performed on Potato Dextrose Agar medium inoculated with standard fungal cell suspensions (1 McFarland) belonging to *A. niger*, *A. flavus*, *A. pseudoglaucus*, *A. nidulans*, *A. flavus oryzae*, *A. versicolor*, *A. montevidensis*, *A. clavatus*, *P. chrysogenum*, *P. corylophylum*, *P. expansum*, *P. digitatum*, *P. brevicompactum*, *P. namylowskii*, *Rhizopus oryzae*, *A. alternata*, *F. proliferatum*, *M. circinelloides*, *P. lilacinum* and on Mueller Hinton (MH) for the bacterial strains, (0.5 McFarland) belonging to *B. subtilis*, *B. megaterium*, *B. thuringiensis*, *B. cereus*, *B. pumilus*, *B. atrophaeus*, *A. aurascens*, *A. globiformis*, and *P. koreensis,* previously isolated from the cultural heritage churches and museum collections of three different geographic locations in Romania [19,20,21]. After 5–7 days of incubation at room temperature/24 h at 37 °C the growth inhibition diameters were measured and converted in arbitrary units as previously described [21].

#### 4.5.2. Quantitative Evaluation of the Antimicrobial Activity of the *R. officinalis* Plant Extract

The antimicrobial activity was performed in Roswell Park Memorial Institute (RPMI, Buffalo, NY, USA) 1640 in the case of fungal strains and in MH Broth medium in the case of bacterial strains, using serial two-fold microdilutions of the extract in 100 μL of broth medium seeded with 1/0.5 McFarland inoculum (corresponding to fungal and bacterial strains, respectively); positive and negative controls were used for each strain. After the incubation period (5–7 days at room temperature/24 h at 37 °C), the minimum inhibitory concentration (MIC) values were established as corresponding to the lowest concentration at which the tested plant extract inhibited the growth of the microbial cultures. The final results were calculated and expressed after the solvent inhibitory effect was eliminated [92].

#### 4.5.3. Influence of *R. officinalis* Hydro-Alcoholic Extract on the Microbial Adherence Capacity to the Inert Substratum

The influence of *R. officinalis* on the microbial adherence ability to the inert substratum (96-well plate, untreated polystyrene) was measured at 490 nm after running the quantitative analysis of the antimicrobial activity through the microtiter method [93]. The minimum biofilm eradication concentration on inert substrate (MBEC) was calculated as follows:(1)$$\mathrm{MBEC}\%=\frac{\mathrm{As}}{\mathrm{Ac}}\times 100$$
where:

As = the absorbance of the microbial adherence treated with *R. officinalis* extract/solvent and Ac = the absorbance of the microbial adherence untreated.

#### 4.5.4. Extracellular NO Release

Quantification of extracellular nitric oxide (NO) release was undertaken for MIC, MIC/2 and MIC/4. NO was rapidly converted to nitrite in aqueous solutions and therefore total nitrite can be used as an indicator of NO concentration. This was measured using a total nitrite spectrophotometric analysis with Griess reagent as previously described with some modifications [65,94]. To the microbial supernatant obtained after 24 h incubation for bacteria and eight days for fungi (50 μL), 50 μL of 2% sulphanilamide in 5% (*v*/*v*) H_3_PO_4_ and 50 μL of 0.13% N-(1-naphthyl)-ethylenediamine aqueous solution were added. Azo dye was measured after 30 min at λ = 540 nm. For the quantification of nitric oxide, a calibration curve with different NaNO_2_ concentrations in the range of 5–100 μM was performed (R^2^ = 0.9998). The results were expressed according to the following equation: (2)$$\%\mathrm{NO}\mathrm{release}=\frac{A\mathrm{sample}-A\mathrm{blank}}{A\mathrm{control}-A\mathrm{blank}}\text{×}100$$
where:

A sample—the absorbance of MIC/2 extract/solvent

A blank—the absorbance of culture media (RPMI or MH broth)

A control—the absorbance of culture media inoculated

#### 4.5.5. The Modulation of Enzymatic Activity by *R. officinalis* Hydro-Alcoholic Extract

Microbial strains grown treated with subinhibitory concentration of hydro-alcoholic extract (MIC/4) and solvent control were evaluated for their capacity to secrete organic acids and enzymes involved in the biodegradation process (cellulase, phenoloxidase, esterase, caseinase and amylase) as previously described [21,81,95,96]. Briefly, different culture media Sabouraud (Sab) supplemented with cellulose and Congo red for cellulase production was highlighted as a clear zone around the colonies; Sab malt agar with tannic acid for phenoloxidase activity was revealed as a brown zone on the colony’s reverse; Sab peptone agar with Tween 20/Tween 80 for esterase detection was highlighted as a precipitation zone on the colony’s reverse; Sab agar supplemented with milk for caseinase detection was revealed by a white-yellow precipitate of calcium para-caseinate around the colonies; Sab agar supplemented with rice starch and hydrolysis was highlighted by flooding the plate with Lugol solution (yellow ring around the culture spot, while the rest of the medium turns blue); and Sab agar supplemented with sodium nitrate, dipotassium phosphate, potassium chloride, glucose, and phenol red for acid production was revealed by a colour change from red to yellow near the inoculum that was spotted with 10 µL suspension adjusted to 1/0.5 McFarland (fungal/bacterial strains) prepared from each microbial strain (untreated culture, serving as growth control and treated with *R. officinalis*) and then incubated for 5–8 days at 26–28 °C.

The influence of *R. officinalis* extract and solvent control on cellulase, phenoloxidase, esterase, caseinase, amylase and organic acid production was evaluated by using the following relation: (3)$$\mathrm{Inhibition}(\%)=\frac{D2-C2}{D1-C1}\times 100,$$
where:

C1—colony diameter of strain control,

D1—clear/brown/white/yellow precipitate/ring surrounding the culture spot/yellow zone diameter of strain control,

C2—colony diameter of sample,

D2—clear/brown/precipitation/yellow zone diameter of sample.

### 4.6. Statistical Analysis

Data were expressed as means ± SD determined by triplicate analysis. The statistical analysis was conducted using GraphPad Prism 9. Data were analysed using ordinary two-way ANOVA with two-stage linear step-up procedure of Benjamini, Krieger and Yekutieli, with individual variances computed for comparison between extract and solvent biological activities. The level of significance was set to *p* < 0.05.

## 5. Conclusions

To the best of our knowledge, this is the first study conducted to demonstrate the efficiency of *R. officinalis* extract against deteriogenic strains isolated from the surface of cultural heritage objects and buildings. The antimicrobial activity of *R. officinalis* extract differs depending on the species and substrate, the best results being obtained for the Gram-positive bacterial strains and for the *A. clavatus* fungal species. The antimicrobial activity of *R. officinalis* hydro-alcoholic extract against fungal and bacterial strain was associated with the presence of hesperidin, vanillic acid and quercetin. The *R. officinalis* extract exhibited anti-biofilm activity, correlating with the release of the NO stress signalling molecule against *P. chrysogenum* strains isolated from wooden or textile materials or bacteria belonging to *B. thuringiensis* species. The inhibitory concentrations of extract or solvent modulated the enzymatic and organic acids profile of the tested biodeteriogenic strains. Taken together, these results recommend the *R. officinalis* hydro-alcoholic extract for further studies (with an emphasis on determining its impact on the heritage object itself) for the development of natural biocides used in cultural heritage conservation.

## Figures and Tables

**Figure 1 ijms-23-11463-f001:**
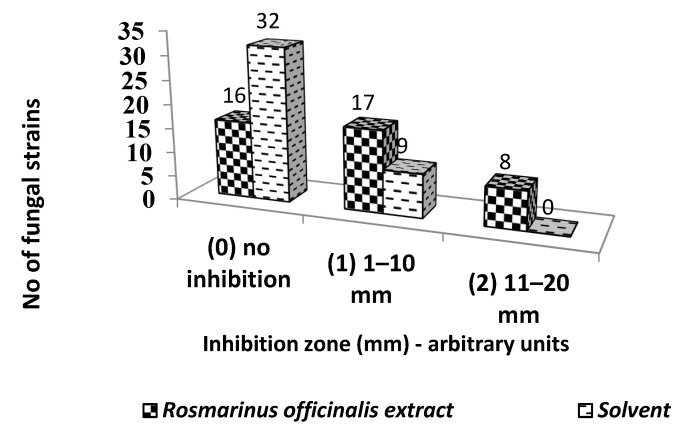
Graphic representation of the *R. officinalis* inhibition arbitrary units against fungal strains.

**Figure 2 ijms-23-11463-f002:**
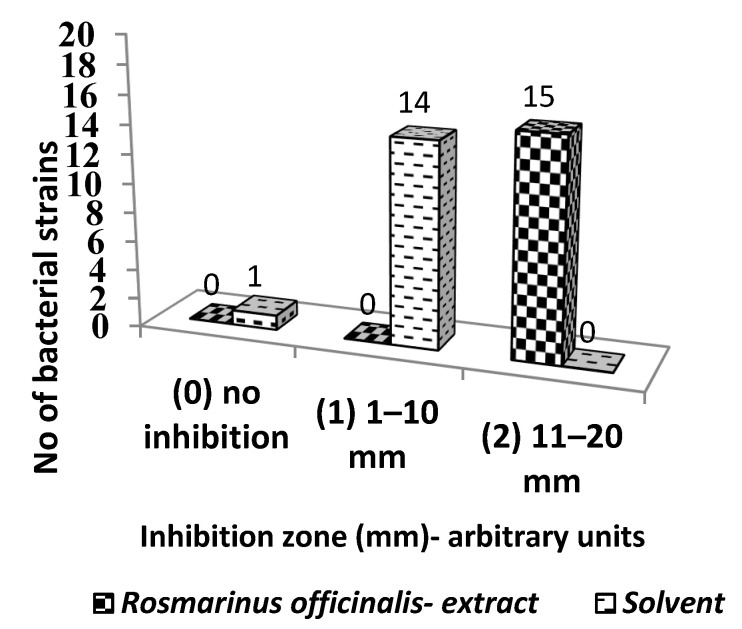
Graphic representation of the *R. officinalis* inhibition arbitrary units against bacterial strains.

**Figure 3 ijms-23-11463-f003:**
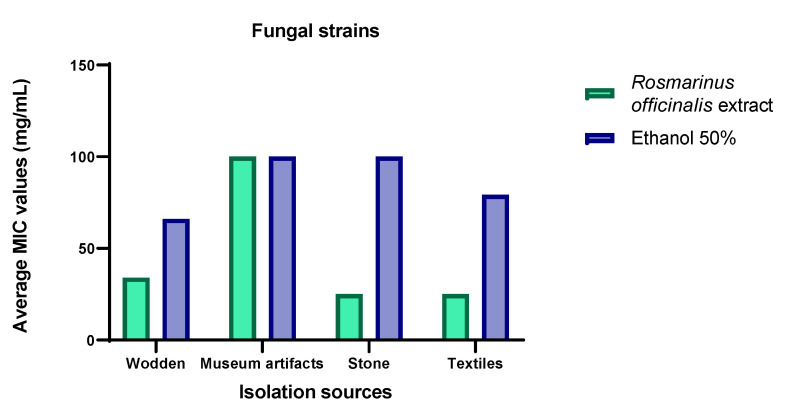
Graphic representation of the average values of the minimal inhibitory concentration (MIC) of *R. officinalis* hydro-alcoholic extract against fungal strains expressed by the isolation sources.

**Figure 4 ijms-23-11463-f004:**
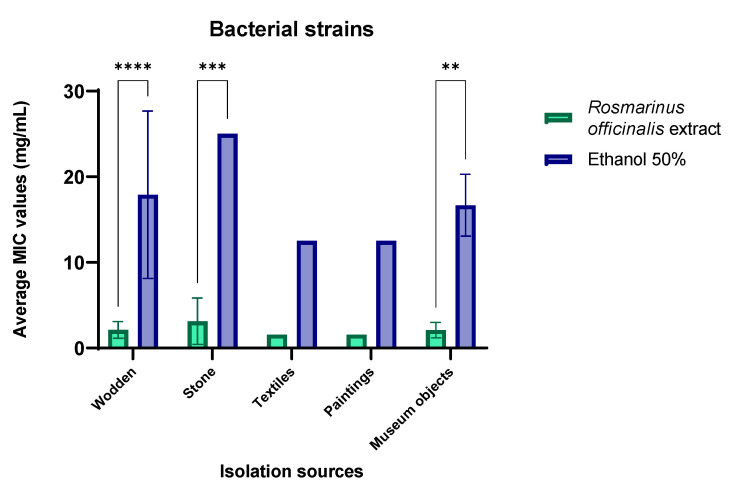
Graphic representation of the average values of the minimal inhibitory concentration (MIC) of *R. officinalis* hydro-alcoholic extract against bacterial strains expressed by the isolation sources. (** *p* < 0.01, *** *p* < 0.001, **** *p* < 0.0001).

**Figure 5 ijms-23-11463-f005:**
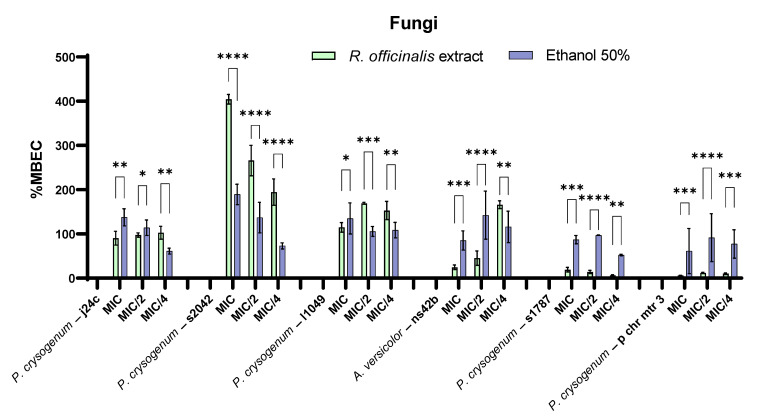
MBEC (%) values for *R. officinalis* extract and corresponding solvent concentration against fungal biodeterioration strains (* *p* < 0.05, ** *p* < 0.01, *** *p* < 0.001, **** *p* < 0.0001).

**Figure 6 ijms-23-11463-f006:**
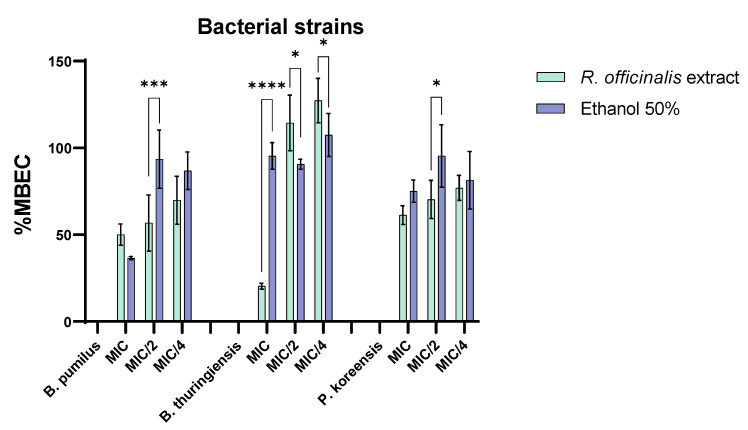
MBEC (%) values for *R. officinalis* extract and corresponding solvent concentration against bacterial biodeterioration strains (* *p* < 0.05, *** *p* < 0.001, **** *p* < 0.0001).

**Table 1 ijms-23-11463-t001:** The TPC and antioxidant activity obtained by different methods.

Parameter	*R. officinalis* Extract	
TPC (mg GAE/mL)	15.62 ± 0.97	
TFC (mg QE/mL)	1.33 ± 0.01	
**% TFC from TPC**	**8.53**	
RSC (mg/mL)	5.97 ± 0.48	
**Antioxidant Activity**	** *R. officinalis* ** **Extract** **(mM Trolox/mL)**	**Ferulic Acid** **(mM Trolox/mg)**
DPPH	18.34 ± 0.18	1.29 ± 0.07
CUPRAC	22.46 ± 0.89	2.59 ± 0.10
FRAP	19.56 ± 2.20	3.64 ± 0.27
TEAC	24.24 ± 0.93	2.04 ± 0.22

**Table 2 ijms-23-11463-t002:** Identification and quantitative data of phenolic compounds in the *R. officinalis* extract using extract UHPLC-MS/MS in negative ionization mode.

No	Compound	Retention Time [min]	Accurate Mass [M-H]^−^	Mass Fragments	Concentration (µg/L)
1	Gallic acid	2.07	169.0133	125.0231	27.98
2	3,4-Dihydroxybenzoic acid	4.35	153.0182	109.0281	301.63
3	Syringic acid	4.02	197.0446	182.0212, 166.9976, 153.0547, 138.0311, 123.0075	836.12
4	4-Hydroxybenzoic acid	6.60	137.0232	93.0331	232.80
5	Vanillic acid	6.75	167.0341	152.0105,124.0154, 111.0075,139.0025, 95.0125	7512.58
6	Chlorogenic acid	6.95	353.0880	191.0553	999.51
7	p-Coumaric acid	7.82	163.0390	119.0489	190.83
8	Caffeic acid	8.04	179.0340	135.044	1637.63
9	Ellagic acid	8.92	300.9989	300.9990	291.86
10	t-Ferulic acid	8.94	193.0498	178.0262, 134.0361	234.67
11	Cinnamic acid	10.45	147.0441	119.0489, 103.0387	611.80
12	Abscisic acid	15.08	263.1286	179.9803, 191.9454	5.75
**Ʃ** **phenolic acids (mg/L)**					**12.88**
13	Catechin	6.47	289.0719	109.0282, 125.0232, 137.0232, 151.0390, 203.0708, 245.0817	33.33
14	Epi-catechin	7.57			664.45
15	Myricetin	8.65	317.0303	178.9986, 164.9263, 151.0036, 137.0244, 107.0125	55.58
16	Rutin	8.98	609.1465	609.1460, 301.0349, 300.0274	Nd *
17	Naringin	9.25	579.1718	363.0721	71.33
18	Hesperidin	9.33	609.1826	377.0876	10,389.09
19	Quercetin	10.74	301.0356	151.0226, 178.9977, 121.0282, 107.0125	70.94
20	Isorhamnetin	11.13	315.0512	300.0276	121.59
21	Kaempferol	11.62	285.0406	151.0389, 117.0180	29.30
22	Apigenin	11.68	269.0455	117.0333, 151.0027, 107.0126	197.76
23	Pinocembrin	12.61	255.0662	213.0551, 151.0026, 107.0125	64.47
24	Chrysin	13.38	253.0503	143.0491, 145.0284, 107.0125, 209.0603, 63.0226, 65.0019	61.357
25	Galangin	13.54	269.0455	169.0650, 143.0491	86.98
**Ʃ** **flavonoids (mg/L)**					**18.88**
26	t-Resveratrol	9.55	227.0707	185.0813, 143.0337	1.14
27	Caffeic Acid Phenethyl Ester	13.17	283.0976	174.9551, 112.9843	54.03

* Nd: not detected.

**Table 3 ijms-23-11463-t003:** The chemical compounds identified in the *R. officinalis* extract by UHPLC-Q-Exactive high-accuracy analysis of deprotonated precursors and fragment ions of specific components combined with data processing using Compound Discoverer software.

Compound Name	Formula	R.T. ^a^ (min)	Exact Mass	Accurate Mass, [M-H]^−^	Experimental Adduct Ion (m/z)	MS^2^ Fragments (m/z)
1-alpha-D-Galactosyl- myo-inositol	C_12_H_22_O_11_	0.74	342.11621	341.1084	341.1088	191.0553, 89.0230, 71.0124
3-Methoxy-4-hydroxy phenylglycolglucuronide	C_15_H_20_O_10_	6.54	360.10565	359.0979	359.0985	197.0448, 179.0340, 135.0439
1-O-coumaroyl-beta-D-glucose	C_15_H_18_O_8_	7.82	326.10017	325.0924	325.0930	119.0489, 135.0439, 161.0233, 179.0341
Tuberonic acid glucoside	C_18_H_28_O_9_	8.28	388.17333	387.1655	387.1661	161.0233, 101.0230, 207.1019
1-O-Sinapoyl-beta-D-glucose	C_17_H_22_O_10_	8.49	386.1213	385.1135	385.1142	59.0124, 71.0124, 121.0282, 161.0234
Bracteatin-6-O-glucoside	C_21_H_20_O_12_	9.00	464.09548	463.0877	463.0883	301.0352
Luteolin 7-rutinoside	C_27_H_30_O_15_	9.32	594.1584	593.1511	593.1514	135.0440, 197.0449, 420.1880
Luteolin 7-glucoside	C_21_H_20_O_11_	9.37	448.1005	447.0932	447.0932	285.0403, 315.0721, 101.0230, 103.0368
Neohesperidin	C_28_H_34_O_15_	9.37	610.18977	609.1820	609.1824	301.0717
Isorhamnetin-rutinoside	C_28_H_32_O_16_	9.42	624.169	623.1617	623.1617	315, 300, 285, 241, 199
Isorhamnetin 3-glucoside	C_22_H_22_O_12_	9.51	478.1111	477.1038	477.1035	315.0510, 299.0196
Rosmarinic acid	C_18_H_16_O_8_	9.54	360.0845	359.0772	359.0770	315.0511, 299.0196, 161.0233
Apigenin-7-O-rutinoside	C_27_H_30_O_14_	9.69	578.1635	577.1562	577.1564	161.0235, 197.045, 135.0440, 269.0457
Apigenin 7-glucoside	C_21_H_20_O_10_	9.80	432.1056	431.0983	431.0984	161.0234, 315.0512, 268.0378, 298.0483
Hispidulin-rutinoside (diosmin)	C_28_H_32_O_15_	9.89	608.1741	607.1668	607.1671	299.0563, 197.0449
Homoplantaginin (Hispidulin 7-glucoside)	C_22_H_22_O_11_	9.93	462.1162	461.1089	461.1091	283.0248
Isoscoparine	C_22_H_22_O_11_	9.95	462.11621	461.1084	461.1091	283.0248, 286.0443
(-)-Usnic acid	C_18_H_16_O_7_	10.00	344.0896	343.0818	343.0823	74.0233, 96.9587, 135.0440, 161.0234
Methyl rosmarinate	C_19_H_18_O_8_	10.06	374.1001	373.0928	373.0929	179, 135, 305
Luteolin 7-glucuronide	C_21_H_18_O_12_	10.39	462.0798	461.0725	461.0727	313.0720, 285.0405, 161.0234, 135.0440
(6aS_11aS)-3_6a_9-Trihydroxypterocarpan	C_15_H_12_O_5_	10.70	272.06847	271.0607	271.0613	96.9587, 133.0283
Luteolin 3′-(4″-acetylglucuronide)	C_23_H_20_O_13_	10.77	504.0903	503.0830	503.0833	285.0405
Rosmanol isomers	C_20_H_26_O_5_	11.09/11.49/13.09	346.178	345.1707	345.1706	283.1702
19-Oxoandrost-4-ene-3_17-dione	C_19_H_24_O_3_	11.76	300.17254	299.1647	299.1651	283.1703, 243.1023, 96.9587
5,6,7-Trihydroxy-4′-methoxyflavone	C_16_H_12_O_6_	11.90	300.0633	299.0560	299.0560	96.9584, 284.0326
(-)-Sophorol isomers	C_16_H_12_O_6_	11.92/12.99	300.06339	299.0556	299.0560	284.0326, 285.0360
(-)-bisdechlorogeodin	C_17_H_14_O_7_	11.69	330.07395	329.0662	329.0667	283.1702, 284.1736
Carnosic Acid	C_20_H_28_O_4_	12.28	332.1987	331.1914	331.1915	283.1703, 301.1804
Genkwanin	C_16_H_12_O_5_	14.00	284.0684	283.0611	283.0610	268.0376
Cortisone	C_21_H_28_O_5_	14.13	360.19367	359.1859	359.1865	317.1758, 283.1703, 96.9587
Rosmaquinone isomers	C_20_H_24_O_5_	14.28/15.36	344.1623	343.1550	343.1549	299.1651, 315.1602, 283.1702, 187.1652
Carnosol	C_20_H_26_O_4_	14.88	330.1831	329.1758	329.1757	285.1859
10_16-Dihydroxyhexadecanoic acid	C_16_H_32_O_4_	15.36	288.23006	287.2223	287.2227	96.9587, 155. 1430
21-Deoxycortisol	C_21_H_30_O_4_	17.97	346.21441	345.2066	345.2069	96.9587, 183.0112
Carnosic Acid 12-Methyl Ether	C_21_H_30_O_4_	17.97	346.2144	345.2071	345.2070	286.1937
Oleanolc acid	C_30_H_48_O_3_	19.27	456.36035	455.3526	455.3530	337.2054, 340.2030
Ursolic acid	C_30_H_48_O_3_	19.43	456.36035	455.3526	455.3530	381.2316, 339.1997

^a^ R.T.—retention time.

**Table 4 ijms-23-11463-t004:** Comparative extracellular NO release (%) between *R. officinalis* extract and similar ethanol concentrations (mean ± SD).

Species	Isolation Source	Concentration (mg/mL)	NO Release (%)		
			*R. officinalis* Extract	Ethanol 50%	*p*-Value
**Fungal strains**					
*P. chrysogenum*	textiles	MIC	1269.81 ± 113.21	2043.40 ± 65.36	<0.001
		MIC/2	Bdl *	Bdl *	-
		MIC/4	Bdl *	Bdl *	-
*A. versicolor*	stone church	MIC	Bdl *	269.81 ± 32.68	<0.001
		MIC/2	6949.06 ± 214.30	Bdl *	<0.000001
		MIC/4	1609.43 ± 130.72	Bdl *	<0.0001
**Bacterial strains**					
*B. thuringiensis*	stone church	MIC	Bdl *	103.11 ± 2.69	<0.000001
		MIC/2	Bdl *	120.22 ± 31.06	<0.01
		MIC/4	127.99 ± 14.25	149.77 ± 7.13	>0.05
*P. koreensis*	stone church	MIC	Bdl *	Bdl *	-
		MIC/2	53.49 ± 40.28	40.23 ± 69.77	>0.05
		MIC/4	Bdl *	169.77 ± 40.28	<0.01

* Bdl—below detection limit (LOD = 1.686 µM).

**Table 5 ijms-23-11463-t005:** (**a**) The influence of *R. officinalis* extract and ethanol (50%) control on the production of enzymatic factors involved in biodeterioration. (**b**) The influence of *R. officinalis* extract and ethanol (50%) control on the organic acid production.

(a)										
Species	Isolation Source	Cellulase Activity (%)			Caseinase Activity (%)			Amylase Activity (%)		
		*R. officinalis* Extract	Ethanol 50%	*p*-Value	*R. officinalis* Extract	Ethanol 50%	*p*-Value	*R. officinalis* Extract	Ethanol 50%	*p*-Value
**Fungal strains**										
*A. niger*	wooden church	-	-	-	106.25 ± 14.32	103.13 ± 0.00	0.0891	-	-	-
*P. digitatum*	museum objects	-	-	-	105.72 ± 4.95	117.14 ± 13.09	0.7756	118.18 ± 7.87	109.09 ± 0.00	0.4070
*A. montevidensis*	textiles	47.44 ± 2.34	42.04 ± 4.05	0.6219	106.82 ± 10.41	106.82 ± 7.87	0.2980	83.33 ± 7.22	108.33 ± 19.09	0.0242
*P. chrysogenum*	textiles	-	-	-	50.00 ± 30.30	78.57 ± 10.10	>0.9999	161.11 ± 7.86	138.89 ± 23.57	0.0446
*A. nidulans*	stone church	-	-	-	83.33 ± 7.86	105.56 ± 7.86	0.0103	2.5 ± 4.33	101.25 ± 5.30	<0.0001
*P. chrysogenum*	textiles	156.25 ± 5.41	168.75 ± 0.00	0.2549	100.00 ± 23.57	111.11 ± 9.62	0.0445	77.19 ± 10.96	0.00	<0.0001
*P. chrysogenum*	museum objects	102.27 ± 6.82	90.91 ± 15.75	0.3006	-	-	-	-	-	-
*A. versicolor*	stone church	125.68 ± 5.73	85.14 ± 17.20	0.0003	-	-	-	-	-	-
*A. nidulans*	stone church	202.5 ± 10.61	243.75 ± 0.00	0.4936	-	-	-	-	-	-
*P. expansum*	museum objects	-	-	-	-	-	-	-	-	-
*P. chrysogenum*	wooden church	-	-	-	-	-	-	100.00 ± 0.00	111.11 ± 9.62	0.3113
*P. namyslowskii*	wooden church	-	-	-	-	-	-	-	-	-
*Aspergillus* spp.	stone church	-	-	-	-	-	-	-	-	-
**Bacterial Strains**										
**Species**	**Isolation Source**	**Caseinase Activity (%)**			**Amylase Activity (%)**			**Esterase Activity (%)**		
		** *R. officinalis* ** **Extract**	**Ethanol 50%**	** *p* ** **-Value**	** *R. officinalis * ** **Extract**	**Ethanol 50%**	** *p* ** **-Value**	** *R. officinalis * ** **Extract**	**Ethanol 50%**	** *p* ** **-Value**
*B. subtilis*	museum objects	97.67 ± 6.98	100.00 ± 4.03	0.8648	108.57 ± 4.95	100.00 ± 17.84	0.5311	-	-	-
*B. cereus*	museum objects	121.43 ± 12.37	110.71 ± 6.19	0.4342	175.00 ± 25.00	175.00 ± 0.00	>0.9999	100.00 ± 26.65	130.77 ± 48.04	0.0278
*B. thuringiensis*	stone church	95.00 ± 4.33	95.00 ± 11.46	>0.9999	114.29 ± 20.20	114.29 ± 24.74	>0.9999	123.08 ± 13.32	107.69 ± 13.32	0.2629
*B. cereus*	museum objects	-	-	-	-	-	-	133.33 ± 14.43	137.50 ± 17.68	0.7604
*P. koreensis*	stone church	102.27 ± 6.82	102.27 ± 6.82	>0.9999	-	-	-	-	-	-
*B. megaterium*	wooden church	-	-	-	80.95 ± 8.25	100.00 ± 14.29	0.1671	-	-	-
*B. megaterium*	museum objects	100.00 ± 4.68	100.00 ± 4.68	>0.9999	57.14 ± 20.20	78.57 ± 10.10	0.1210	-	-	-
**(b)**										
**Species**	**Isolation Source**	**Organic Acid—Fungal Strains**								
		** *R. officinalis * ** **Extract**	**Ethanol 50%**	** *p* ** **-Value**						
*A. niger*	wooden church	110.00 ± 18.66	100.00 ± 17.32	0.3618						
*P. digitatum*	museum objects	108.70 ± 7.53	117.39 ± 0.00	0.4279						
*A. montevidensis.*	textiles	-	-	-						
*P. chrysogenum*	textiles	-	-	-						
*A. nidulans*	stone church	-	-	-						
*P. chrysogenum*	textiles	-	-	-						
*P. chrysogenum*	museum objects	-	-	-						
*A. versicolor*	stone chuch	116.67 ± 7.22	122.5 ± 0.00	0.7032						
*A. nidulans*	stone church	98.08 ± 24.48	96.15 ± 6.66	0.8600						
*P. expansum*	museum objects	-	-	-						
*P. chrysogenum*	wooden church	104.76 ± 8.25	147.62 ± 32.99	0.0002						
*P. namyslowskii*	wooden church	-	-	-						
*Aspergillus* spp.	stone church	-	-	-						

“-” refers to not tested strains in presence of extract or solvent.

## Data Availability

Samples of the compounds are available from the authors.

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
