# Peer review of "Eco-Friendly Solution Based on Rosmarinus officinalis Hydro-Alcoholic Extract to Prevent Biodeterioration of Cultural Heritage Objects and Buildings"

_ijms, 2022, doi:10.3390/ijms231911463_

Round 1

Reviewer 1 Report

The article provides interesting information about a green chemical product extracted from rosmarinus officinalis. The efficacy of the product has been tested with a diverse and adequate methodology and the conclusions are well related to what is stated in the text. I recommend writing in italics the names of the genera and species mentioned and unifying the abbreviations. In some cases abbreviation is used in the genre and in others not.  

Author Response

Dear reviewer,

Thank you very much for all your valuable suggestions that have greatly helped us to improve the manuscript. We tried to answer your pertinent and useful observations. We marked our changes using the yellow color in the manuscript. Our responses are listed below (red).

Reviewer 2 Report

The aim of the reviewed work was to evaluate the hydro-alcoholic extract of Rosmarinus officinalis to inhibit growth and biodeteriorating potential fungi and bacteria isolated from cultural property items. There are presented in the work numerous data dealing the chemical composition of the extract, the antimicrobial activity, antioxidant activity etc. In the reviewer’s opinion this goal has been achieved sufficiently.

But the primary target of research in this field is the practical application of the proposed agent for the cultural heritage items. I would like to read in the “Discussion” chapter the Authors’ opinion about the potential use of the hydro-alcoholic extract of rosemary for historic supports – possibly for stone, wood, canvas, paper?... Furthermore, the practical application of a new biocide requires many investigations of its influence on these supports. The biocide proposed to the conservation of cultural heritage should be not only eco-friendly but it has to be first of all effective against the harmful microorganisms and safe for the historic matter. Moreover, it has to be allowed to use in the European Union legally.

In the reviewer’s opinion the Authors should take into consideration publications of other researchers dealing biocidal essential oils from various plants, including works of Rakotonirainy et al. (IBB 55, 2005, 141-147), Matusiak et al. (IBB 131, 2018, 88-96), Lavin et al. (Microb. Ecol.  71, 2016, 628-633) and other.

Why the “Materials and Methods” chapter follows “Results” and “Discussion”? In the reviewer’s opinion it is less comfortable for the reader than the traditional order: Introduction, Materials and Methods, Results, Discussion, Conclusions.

Author Response

(The authors gave the same response as above.)

Reviewer 3 Report

The manuscript "Eco-friendly solution based on Rosmarinus officinalis hydro-al-coholic extract to prevent biodeterioration of cultural heritage objects and buildings" by Corbu et al. presents interesting results. Before it can be accepted for publication, some points should be addressed by the authors, in my opinion:

1. Importance of cultural heritage should be underlined, so as the effects of fungi on CH objects

2. How were the extraction procedure parameters selected? Please present also the reason for selecting the hydro-alcoholic mixture as solvent.

3. What is the importance of phytochemical assays (TPC, TFC, RSC) for the final application? This should be clearly presented, otherwise those results does not present any significance for the present work. Similar discussion for antioxidant assays.

4. Why are those specific lines selected for the antimicrobial lines? Where any identified on CH objects by the authors or the selection is based on literature data? How they affect CH objects?

5. Results - The Rosmarinic acid is well-known as an antioxidant and antimicrobial agent (among other properties). How is relevant for the present work the antioxidant properties exhibited by the extract?

6. Antioxidant assays - in my opinion, the comparison between different assays is not very practical. The authors should use a positive control (i.e., rosmarinic acid). Also, if possible, the IC50 should be presented.

7.   A positive control should be used for the antimicrobial assays

8. The authors propose the application of natural extracts as potent antimicrobial agents for application on a series of CH objects. In my opinion, at least in the conclusions section, it should be underlined that this proposal should also carefully consider the potential damages induced on the objects (how an extract could affect a textile or wood? what aestethic changes could be induced? etc). As the work addresses the conservation of CH objects, the possible concerns of restorers should also be taken into consideration. 

9. The manuscript contains some typos, grammatical errors and some track-changes marks. Please carefully check the manuscript.

Author Response

(The authors gave the same response as above.)

Round 2

Reviewer 3 Report

The authors addressed the reviewer's remarks in a satisfactory manner. More important, the authors performed the necessary corrections as such that the manuscript can be understood by the general public interested in this area of research. Preservation of cultural heritage remains an important research area, as such any new data is welcomed. Some minor errors (such as inconsistent format in the references section) can be resolved during the final editing steps. The manuscript can be accepted for publication in the present form.